# The Effects of Electrolytic Technology Toothbrush Application on the Clinical Parameters and Bacteria Associated with Periodontal Disease in Dogs

**DOI:** 10.3390/ani14213067

**Published:** 2024-10-24

**Authors:** Nemanja Zdravković, Nemanja Stanisavljević, Milka Malešević, Goran Vukotić, Tatjana Stevanović, Ivan Bošnjak, Milan Ninković

**Affiliations:** 1Scientific Institute of Veterinary Medicine of Serbia, Janisa Janulisa 14, 11000 Belgrade, Serbia; milan.ninkovic1992@gmail.com; 2Institute of Molecular Genetics and Genetic Engineering, Vojvode Stepe 444a, 11042 Belgrade, Serbia; nemanjalab08@imgge.bg.ac.rs (N.S.); milkam@imgge.bg.ac.rs (M.M.); vukoticg@bio.bg.ac.rs (G.V.); 3Faculty of Biology, University of Belgrade, Studentski trg 16, 11158 Belgrade, Serbia; 4Veterinary Clinic “Pas, Mačka i …”, Omladinskih brigada 7đ, 11070 Belgrade, Serbia; pasmackacom@gmail.com; 5Academy of Applied Preschool Teaching and Health Studies, Balkanska 18, 37000 Krusevac, Serbia; ivanbosnjak74@gmail.com

**Keywords:** bacteria, canine, clinical score, dental, stomatology, quality of life, periodontal disease

## Abstract

**Simple Summary:**

There is a lack of data on the usage of electrolytic toothbrushes and their effects on oral health in dogs. The aim of this study was to investigate the beneficial effects of electrolytic toothbrushing compared to mechanical toothbrushing in everyday routines for dogs. The daily usage of an electrolytic toothbrush improves oral health clinical parameters and reduces the abundance of some periodontal-disease-associated bacteria in dogs. The encouraging results support the usage of electrolytic toothbrushes in routine oral hygiene practice for dogs.

**Abstract:**

The aim of this study was to compare the effects of electrolytic and nonelectrolytic toothbrushing on dogs’ oral health and the presence of common bacteria associated with periodontal disease. Periodontal disease in dogs is a common problem worldwide. A toothbrushing procedure is recommended to prevent periodontal disease, with additional benefits if electrolytic toothbrushes are used in dog oral hygiene practices. A total of 26 dogs were enrolled in this eight-week study and were divided into two groups—treatment and control. Daily toothbrushing was performed on all dogs using the same dog toothbrush, with the power source disengaged in the control group. Oral examination was conducted on anesthetized dogs before and at 4 and 8 weeks after commencing the study, with sampling for bacterial analysis. This study was designed to be blind for owners, veterinarians, and laboratory staff. Improvements in the average gingival index (from 0.55 to 0.31) and calculus index (from 0.55 to 0.38) in the treatment group were recorded. In the control group, after an initial improvement in the plaque index (from 0.97 to 0.53), at week 8, it significantly rose to 1.21 (*p* < 0.05). Relative bacterial abundance revealed a reduction in all four tested bacteria in the treatment group, while in the control group, Campylobacter rectus levels rose by 3.67 log2 compared to before the study and at week 8. No adverse effects were recorded in either group.

## 1. Introduction

One of the most common diseases in dogs is periodontal disease [1,2,3]. Periodontal disease affects teeth-supportive tissues [4]. The initial reversible inflammatory stage of periodontal disease is gingivitis, which is clinically presented by swelling and redness; however, the more destructive and irreversible form of periodontal disease is periodontitis, which is characterized by the destruction of the gingival tissue, alveolar bone, cementum, and periodontal ligament [1]. The etiology of periodontal disease in canines is not yet fully understood [5], but it is a complex health problem in which disease expression is associated with interactions between bacteria, their biofilms, and the host’s immunoinflammatory response, as is evident in saliva [6] and subsequent tissue alterations [3]. Although periodontal diseases are quite well known in veterinary medicine, data on their prevalence and risk factors are still scarce [7]. The assessed prevalence of dog periodontal diseases is close to 90% [8]. Periodontal disease has been reported in correlation with general health status concerning renal, hepatic, and cardiac disease, as well as detrimental blood health parameters [5,9].

Several clinical scores are used to assess canine oral health via clinical examination. The state of gingival health is assessed by various indices that are designed to give a numerical value to clinical status [10]. In brief, the gingival index (GI) measures the inflammation level based on the status of the gingival margin and clinical symptoms such as redness, edema, hypertrophy, bleeding, and ulceration.

Bleeding on probing (BOP) evaluates whether the gingival inflammation level is severe enough to induce bleeding upon gentle touch with a dental probe. In aggravated clinical cases, there is spontaneous gum bleeding. The plaque index (PI) is a measure of dental plaque, which is considered a biofilm of oral bacteria made from a matrix of bacteria with extracellular polysaccharides and salivary glycoproteins [8]. If plaque is not removed, it forms a rough surface within a few days, even on clean teeth [11]. The calculus index (CI) represents the dental calculus score, which refers to the mineralized plaque that is not in itself pathological but facilitates the adhesion of dental plaque [5].

There are numerous bacterial species associated with the periodontal disease, but *Eikenella corrodens*, *Fusobacterium nucleatum*, *Parvimonas micra*, and *Campylobacter rectus* are often mentioned in both veterinary and human medicine [1,2,12,13,14,15]. The actual proliferation of Gram-negative anaerobic rods has been strongly implicated in periodontitis pathogenesis [16]. The relative quantification of bacteria permits the possibility of measuring the relative abundance of bacteria and exploring their clinical impact [14].

The primary goal of periodontal treatment is the elimination of or a reduction in pathogenic bacteria, as well as controlling inflammation [3]. It is also considered that changes in the composition of oral microbiota toward increased pathogen abundance, instead of increases in total bacterial number, represent an initial step in the development of periodontal disease [8]. Since the removal of bacterial biofilms is critical for periodontitis treatment [17], this problem is often addressed by the application of various treatments, including topical antiseptics, systemic or local antibiotics, piezoelectric toothbrushing, textile products such as nylon or microfiber, ultrasound devices, and even gaseous ozone or probiotic treatments [2,3,5,11].

To prevent the development of oral pathological conditions, daily toothbrushing is considered a critical home care routine for dogs [8,18,19]. The use of an electrolytic toothbrush should offer additional benefits related to its mode of action. The mechanism of an electrolytic toothbrush involves supplementing the mechanical toothbrushing action with an electrolytic effect on saliva, which changes the polarity of the teeth’s surface, promotes plaque removal, and reduces bacterial adhesion on the tooth surface [20]. Both teeth and plaques are generally negatively charged [20,21]. Although two similarly charged surfaces should electrostatically repel one another, in realistic conditions, plaques stick to teeth because the positively charged calcium (Ca^2+^) ion in the saliva reduces the natural negative charges of both bacteria and the tooth surface, allowing the bacteria and the tooth surface to contact more closely, at which point the van der Waals forces, which are effective at very short distances, allow the bacteria to stick to the teeth [21]. The purpose of the electric component of the electrolytic toothbrush is to form an electric circuit that generates negative ions and, thus, blocks the cross-linking of adhesive molecules on teeth, thereby preventing bacterial attachment [21,22]. The development of dog toothbrushes capable of efficiently and effectively removing dental plaques led to the employment of electrolytic technology, which has previously been recognized in human dentistry [20] to be beneficial in patients with multibracket appliances [23].

Although the first use of an electrolytic electric toothbrush was in 1889 [23], the usage of such devices has not been widely studied in canine preventative oral health. The zoonotic impact of dog periodontal disease is noticeable for pet owners due to similarities between the conditions in dogs and humans; furthermore, the canine animal model is considered strongly applicable in human periodontal disease studies [22,24].

The aim of this study was to investigate whether the use of an electrolytic toothbrush in dogs would be beneficial compared to a nonelectrolytic toothbrush and to observe whether it could be used in their everyday, routine, preventative oral health practice. To conduct this analysis, the dogs’ oral health was examined, and the levels of the most prominent periodontitis-related bacteria were monitored.

## 2. Materials and Methods

### 2.1. Ethical Committee

All study protocols and the involvement of the animals in the experiment were approved by The Veterinary Directorate, Ministry of Agriculture, Forestry and Water Management (No. 323-07-02291/2023-05) and the Animal Research Committee of the Faculty of Veterinary Medicine, University of Belgrade (No. 01-02/2023).

### 2.2. Study Population Inclusion and Exclusion Criteria

This study was focused on house dogs with previous toothbrushing experience. Prior to collection, the owners were informed of the purpose of the present study and gave their consent to participate.

Enrolment criteria included that none of the dogs had received antibiotic treatment in the month prior to the study. No dogs received antibacterial or anti-inflammatory medicines during the study. The dogs were examined by a veterinarian; the study included dogs in a generally healthy condition, with gingival index, plaque index, and bleeding on probing scores no worse than 2. All gingival swab samples were collected at a single veterinary clinic right after a specialist dental examination [25]. The owners were asked to withdraw from the study if they saw any signs of discomfort in their dogs related to the toothbrushing procedure. This study started with 26 dogs. All dogs lived under their respective household conditions, and the owners were asked not to change their feeding practices during the experiment’s duration. The mean age of the animals involved in the study was 4 years and 4 months. Dogs of various breeds participated in the study—information on breed distribution is given in Appendix A.

The experiment was designed as a blind study (neither the owners nor the veterinary dentists or swab examiners knew which group a given dog belonged to). All dogs were randomly distributed into the two groups. Both groups utilized the same toothbrush. The electric power source was disengaged for the control group (C), while the electrolytic treatment group (ET) was treated with a fully functional toothbrush.

### 2.3. Procedure

The experimental period lasted for 8 weeks, with control checks and swab sampling before the experiment and at 4 and 8 weeks. The owner’s involvement included brushing the dogs’ teeth for 2 min once a day—in the evening—for the study’s duration. The owners were instructed to report any adverse or favorable changes in their pet’s behavior during the observation time. An optional observation of the dogs’ oral health was also offered in the two months following the experiment.

### 2.4. Toothbrush Design

Both the control and electrolytic treatment groups used the same model of a dog-specific toothbrush, Petsie (Vishealth, Ltd., Belgrade, Serbia). The toothbrush is made from a rubbery soft substance, allowing the dogs to chew on it, which does not break if bitten. The conductivity is sent through the brush’s bristles directly to the dogs’ mouths. The source of electrical power is a 3 V lithium-ion battery, similar to a watch battery. An electric flow is induced when pressing the button, lasting for 1 min, which is the time estimated for toothbrushing on a single jaw [5,26].

### 2.5. Clinical Examination and Sampling

All the procedures of the dog’s oral examinations were carried out by a veterinary dentist. The clinical assessment score scale is presented in Table 1, following procedures established earlier [5,16] under Zoletil-based anesthesia. Immediately after clinical examination, the sample swabs were taken from the buccal gingiva of both the maxillar and the mandibular sides. All the tubes were processed further in the DNA extraction lab. The total DNA present in each sample was extracted using the phenol/chloroform-based protocol described earlier [27].

### 2.6. qPCR

In order to quantify the specific bacterial populations, the qPCR method was carried out using the MIC qPCR Cycler (Bio Molecular systems, Upper Coomera, Australia) and employing a bacterial intercalating, green-based assay. Each reaction contained 10 µL of Mastermix (FastGene^®®^ IC Green 2 x qPCR Universal Mix, Nippon genetics, Tokyo, Japan), 300 nM of each primer, 1 µg of sampled DNA (the volume ranged from 1 to 5 µL), and PCR-grade water up to a final 20 µL reaction volume. The primers used for quantifying four periodontopathic species previously reported to be associated with periodontitis—*Eikenella corrodens*, *Fusobacterium nucleatum*, *Parvimonas nigra,* and *Campylobacter rectus*—are presented in Table 2. The primers had been previously published [1], with the sensitivity of this method previously reported to be at 100 cells [1,2].

### 2.7. Statistical Analysis

Statistical analyses were performed using the computational software package GraphPad Prism, version 10.0.0 for Windows (GraphPad Software). A comparison was performed using the ANOVA test with Tukey post hoc comparison where applicable. For comparison purposes, the findings at sampling periods 1 and 2 were compared to the baseline findings (sampling time 0) and were analyzed using paired-sample ANOVA, while tests between the C and ET groups were analyzed using ANOVA.

A two-way ANOVA test was performed to examine the source of variation in the test groups regarding each clinical parameter (GI, CI, PI, and BOP) and the abundance of each tested bacterium. The significance level was set at *p* < 0.05.

## 3. Results

### 3.1. Clinical Parameters

Clinical parameters showed a general improvement after 8 weeks of brushing compared to the baseline values. The only statistical difference in the clinical scores was recorded between the plaque indices measured in sampling periods 1 and 2 (*p* = 0.0204); the other values are given in Table 3. The GI showed a reduction in both the C and ET groups.

The CI parameter underwent a temporary increase at 4 weeks in terms of its mean value, but at 8 weeks, the scores dropped below the initial value, indicating an improvement in dental health. The values of PI and BOP showed preliminary reductions during the toothbrushing treatment, as shown in Figure 1.

### 3.2. Bacterial Quantification

The bacterial molecular quantification results are presented in Figure 2. The first sampling period was taken as a baseline for the bacterial abundance comparisons in the two subsequent sampling periods. The bacterial count relative to 10^4^ copy numbers/µL was arbitrarily classified as high. Values between 10^2^ and 10^4^ copy numbers/µL were considered notable, and fewer than 10^2^ copy numbers were considered negative due to the previously reported sensitivity of the method [1,2].

#### 3.2.1. *Eikenella corrodens*

At the beginning of the study, a notable qPCR-detected abundance of *E. corrodens* was found in 23 out of 26 dogs (88.46%), and this prevalence persisted without significant changes in the next two samplings. High molecular abundance was detected in four dogs (15.38%), two from the C group and two from the ET group. In the second sampling period, three dogs had high *E. corrodens* copy numbers, two from the C group and one from the ET group—the same individuals that had high copy numbers in the first sampling. In the last sampling period, no dog exhibited a high copy number of *E. corrodens*.

#### 3.2.2. *Fusobacterium nucleatum*

A notable presence of *F. nucleatum* was discovered in 19 dogs (73.08%) at the baseline examination, with 1 dog (3.85%) having more than 104 copies/µL. A reduction in bacterial abundance at later sample points was observed for this species.

#### 3.2.3. *Parvimonas micra*

The analysis of the presence of *P. micra* showed that eight dogs (30.77%) had notable levels in the baseline period and that this remained unchanged in the two subsequent periods in the same individuals. Only one dog had relatively high copy numbers in the baseline period, and the same individual retained a high copy number in later sampling periods, although the molecular abundance was lower than the baseline level.

#### 3.2.4. *Campylobacter rectus*

The bacterium *C. rectus* was detected in five dogs at the baseline examination (three dogs in the ET group and two dogs in the C group), and this prevalence remained until the end of the experimental period, with little change in individual abundance.

More than one bacterial species with a high copy number was observed in two dogs (7.69%).

The mean differences of Ct values are presented in Table 4. The largest relative change in bacterial copy numbers was found for *C. rectus*, with a mean reduction of over 109-fold between the ET and C groups. The second-best reduction was noted for the same species in the ET group between the baseline and the second sampling period, indicating a near 50-fold reduction in molecular equivalents. In the ET group, the reductions in *E. corrodens*, *F. nucleatum*, and *P. micra* at 8 weeks were greater than those at 4 weeks.

### 3.3. Source of Variation in the Results

To achieve a better understanding of the results of the experiment, two-way ANOVA analysis was conducted to determine whether the clinical parameters or bacterial enumerations were dependent on the type of toothbrush applied or the duration of the treatment. The analysis generally indicated that the main source of variation was the individual oral environment of each observed dog.

#### 3.3.1. Clinical Parameters

The clinical assessment of the oral health parameters in the observed dogs showed that GI and PI scores significantly differed as the period of toothbrushing proceeded. The CI scores were statistically different between the ET and control groups, whereby the ET group showed healthier oral scores throughout the observation period. The results of the variational analysis are given in Table 5.

#### 3.3.2. Bacterial Abundance

The bacterial abundances of the periodontitis-related pathogens *E. corrodens*, *F. nucleatum*, *P. micra*, and *C. rectus* were analyzed in relation to the sampling period, toothbrush type, and individual variations in oral health conditions for each dog. The sampling period has been found to be of statistical significance for *F. nucleatum* and *P. micra,* and the abundances of these two bacteria changed with toothbrushing using any type of toothbrush. The numbers of *E. corrodens* and *P. micra* significantly differed when the ET toothbrush was used compared to the C toothbrush. The sources of variation in bacterial molecular abundance are further shown in Table 6.

## 4. Discussion

This study determined the effects of toothbrushing, based on brushing mechanisms and electrolytic effects, using a novel type of toothbrush on the oral health and prevalence of four periodontal disease-associated bacteria in a healthy, heterogeneous group of dogs without systemic disease. The results show that some of the clinical health parameters, such as the calculus index, benefit from the usage of electrolytic toothbrushes. The calculus index offers information on the level of plaque affected by mineralization [5]; our eight-week study revealed that regular toothbrushing with an electrolytic toothbrush is beneficial for dogs, with significant findings on the upper fourth premolar and first molar teeth, as expected [28]. The gingival index parameter improved over time as long as toothbrushing was performed, while the degree of bleeding on probing remained at the same level, which was expected, as this study was designed for generally healthy dogs with previous toothbrushing experience. Either the use of an electrolytic toothbrush was found to be beneficial or the clinical parameter scores of the dog’s oral health remained the same. No adverse health effects were recorded.

In this study, oral hygiene practices and clinical examinations were performed on dogs with generally good health. The observation protocol revealed minor issues of sensitivity. For example, a dog with deposits of plaque or calculus on one tooth received the same score as a dog with the same amount of deposit on multiple teeth. We found this to be a methodological constraint, so the results might be considered as referring to the worst-case scenario. The amount of deposit was observed, but the positions of the deposits, whether supragingival or subgingival on the individual tooth, were not considered. The mentioned methodological constraints might have led to poorer results in the clinical assessment of PI and BOP at the end of the study period.

The dogs’ CIs improved as a result of electrolytic treatment compared to the control, while GI improvements were more associated with subsequent regular toothbrushing than with the choice of toothbrush. The BOP did not change notably in individual dogs throughout the study. Considering the sources of variation shown by the ET and C groups between the multiple observation time points, the ANOVA showed that dogs’ oral health benefits more from electrolytic toothbrushing than the control treatment in terms of the CI parameter, while favorable changes were produced by subsequent regular toothbrushing in terms of the GI and PI parameters.

These findings once again emphasize toothbrushing as a critical routine oral hygiene practice for canines [5,8], with the introduction of electrolytic technology in toothbrushes as a beneficial supplement.

It is considered that, in studies on periodontal disease, the qPCR method has a higher sensitivity and specificity compared to other microbiological methods, especially in relation to anaerobic pathogens [12]. The changes in the abundance of four bacterial species between the baseline measurement and two additional sampling periods (1 and 2), as well as comparisons between the ET and C groups (in sampling periods 1 and 2), reveal reductions in all bacteria assessed, or in at least one species, in 19 dogs.

The largest and second largest relative changes in bacterial copy numbers between the ET and C groups were shown by *C. rectus*, with a change of over 109-fold, yet more interesting is the finding that other bacteria (*E. corrodens*, *F. nucleatum* and *P. micra*) underwent greater reductions at 8 weeks than at 4 weeks in the ET group, which further confirms the recommendation of regular toothbrushing in dogs. The results derived at 8 weeks from both clinical and molecular analyses were better than those at 4 weeks, and the results at 4 weeks were better than those at the start of the study, indicating that even short-term or irregular toothbrushing would be better than no treatment at all, which is consistent with previous findings [5].

Regarding the rare cases showing an increment in bacterial molecular abundance, these were not found to be either statistically or clinically significant. In the zoonotic context, our findings are supported by those from previous metagenomic analyses of periodontitis-affected dogs’ teeth, with findings showing known periodontal pathogens in both humans and dogs [29].

The optimal design of dog toothbrushes must allow for better access to hard-to-reach areas of the mouth and, thus, be more effective in order to be acceptable for dogs. Although this study focused on dogs with previous toothbrushing experience, a period of at least a month is usually required to train a dog to accept the toothbrushing procedure, which is similar to the 5-week time period reported earlier [5]. It is well known in veterinary dental practice that a significant proportion of dog owners report difficulties in achieving cooperation with their dogs during lengthy toothbrushing procedures, with some even struggling when simply inspecting their dogs’ teeth. In addition, owners are usually unable to recognize signs of dental discomfort in their pets; thus, oral diseases often go unnoticed. Over time, interest in toothbrushing procedures wanes, and subjects often return to their usual toothbrushing practices—a trend even detected in humans [30]. The toothbrushing intentions of dog owners usually relate to complaints of halitosis (bad breath), this being the first recognizable clinical sign associated with periodontal disease [31].

The obtained results are reasonably supportive of the interpretation that even dogs with mild periodontal disease-associated problems may benefit from using an electrolytic toothbrush.

## 5. Conclusions

The dogs’ oral clinical scores either improved or remained the same when an electrolytic toothbrush was used for 8 weeks. No dog showed an adverse reaction to 2 min of daily toothbrushing during the study. Neither the electrolytic treatment group nor the control group showed deteriorating clinical scores.

In most cases, the number of bacteria associated with periodontal disease in dogs was reduced, even in orally healthy dogs.

There is evidence here that supports introducing toothbrushing routines for dogs, with additional benefits to be gained if an electrolytic toothbrush is used.

The findings of this study further support the hypothesis that dogs with mild periodontitis may benefit from the use of an electrolytic toothbrush in addition to regular specialist veterinary dental treatments.

## Figures and Tables

**Figure 1 animals-14-03067-f001:**
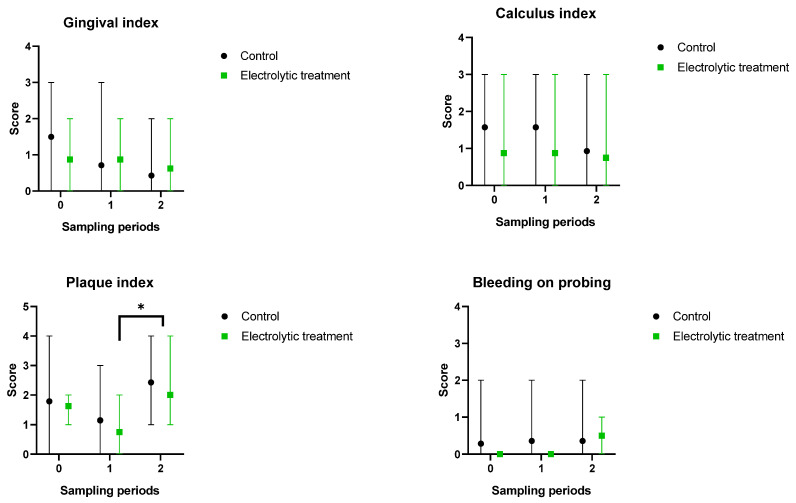
Clinical scores of dogs (mean with range) at the beginning of the study (0) and at 4 (1) and 8 (2) weeks. Gingival and calculus indices showed gradual improvements over time, while bleeding on probing remained at the same level. Asterisk denotes statistical significance (*p* < 0.05).

**Figure 2 animals-14-03067-f002:**
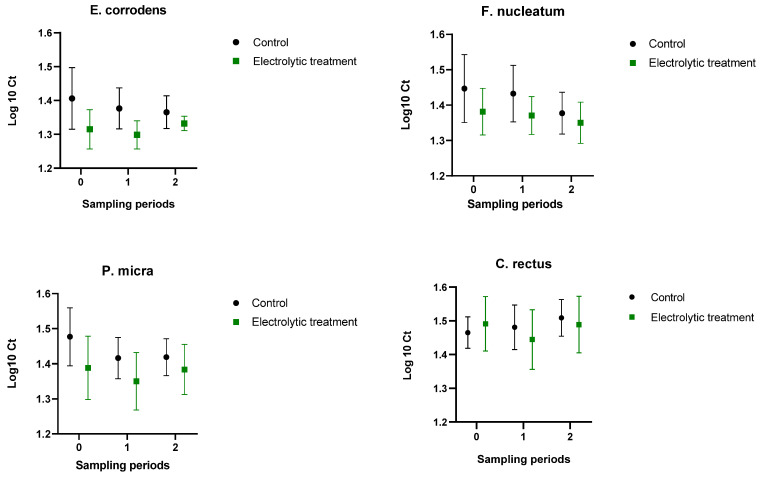
Changes in the bacterial loads of the periodontitis-related bacteria *Eikenella corrodens, Fusobacterium nucleatum, Parvimonas micra,* and *Campylobacter rectus* in dogs at the beginning of the study (0) and at 4 (1) and 8 (2) weeks. Although not statistically significant, there was a mild reduction in periodontitis-related bacteria in the electrolytic toothbrush group, while in the control group, the *C. rectus* load was mildly increased.

**Table 1 animals-14-03067-t001:** The clinical oral health parameters of plaque, gingival, and calculus indices, as well as bleeding on probing, were scored using the following scales at the beginning of the study and at 4 and 8 weeks.

	Score	Grade	Reference
Plaque index (PI)	0	No plaque	[16]
	1	A film of plaque adhering to the free gingival margin and adjacent area of the tooth (not more than 1 mm)	
	2	Can be seen with the naked eye (less than 1/2 of crown)	
	3	Abundance of soft matter within the gingival pocket and/or on the tooth and gingival margin (more than 1/2 of crown)	
Gingival index (GI)	0	Absence of inflammation	[16]
	1	Mild inflammation—slight change in color of gingival margin and little change in texture	
	2	Moderate inflammation—moderate glazing, redness, edema, and hypertrophy; bleeding on pressure	
	3	Severe inflammation—marked redness and hypertrophy, spontaneous bleeding and ulceration	
Calculus index (CI)	0	No calculus	[5]
	1	Supragingival or calculus that extends only slightly below the free gingival margin	
	2	Moderate amount of supra- and/or subgingival calculus or only subgingival calculus	
	3	Abundant supragingival and/or subgingival calculus	
Bleeding on probing (BOP)	0	Absence of bleeding within 10 s following probing	[16]
	1	Presence of bleeding within 10 s following probing	

**Table 2 animals-14-03067-t002:** The primers used for qPCR at the beginning of the study and at 4 and 8 weeks for the analysis of the relative abundance of targeted periodontal disease-associated pathogenic bacteria, such as *Eikenella corrodens*, *Fusobacterium nucleatum*, *Parvimonas micra*, and *Campylobacter rectus*.

Primer Sequence 5′–3′	Primer	Amplicon Product Size
GCGAAGTAGTGAGCGAAGAG	Campy.rectus F	119
GCCTGCGCCATTTACGATA	Campy.rectus R	
AGGCGACGAAGGACGTGTAA	Eikk.corr F	69
ATCACCGGATCAAAGCTCTATTG	Eikk.corr R	
GACATCTTAGGAATGAGACAGAGATG	Fus.nucle F	73
CAGCCATGCACCACCTGTCT	Fus.nucle R	
AAACGACGATTAATACCACATGAGAC	Par.micr F	201
ACTGCTGCCTCCCGTAGGA	Par.micr R	
GCAAGAACGTGATGACGGGA	Prev.nig F	79
ATTTCGCAGTCTTTGGGATCTTT	Prev.nig R	

**Table 3 animals-14-03067-t003:** Oral clinical parameters scores (mean ± standard deviation) for 26 dogs in two study groups (control and electrolytic treatment) at the beginning of the study (0) and at 4 (1) and 8 (2) weeks.

	Sampling Period	Control	Electrolytic Treatment
Gingival index	0	0.72 ± 0.55	0.55 ± 0.44
	1	0.33 ± 0.59	0.44 ± 0.39
	2	0.21 ± 0.32	0.31 ± 0.46
Calculus index	0	0.75 ± 0.45	0.55 ± 0.55
	1	0.83 ± 0.41	0.50 ± 0.56
	2	0.54 ± 0.6	0.38 ± 0.58
Plaque index	0	0.97 ± 0.62	0.95 ± 0.95
	1	0.53 ± 0.58 ^a^	0.44 ± 0.44
	2	1.21 ± 0.43 ^a^	1 ± 1
Bleeding on probing	0	0.16 ± 0.35	0 ± 0
	1	0.17 ± 0.36	0 ± 0
	2	0.18 ± 0.32	0.25 ± 0.25

Statistical differences between the same letters (^a^ *p* < 0.05).

**Table 4 animals-14-03067-t004:** Fold changes in the bacterial numbers (∆Ct values) and the log2 changes in relative bacterial abundance. Results from sampling periods 1 (4 weeks) and 2 (8 weeks) are compared to the reference (sampling period 0), as well as between the control and electrolytic treatments. Negative numbers reveal a reduction. The largest difference between the C and ET groups was found in *C. rectus* levels after 8 weeks of study. Electrolytic treatment led to larger reductions in *E. corrodens*, *F. nucleatum*, and *C. rectus* compared to the control group in both sampling periods.

	Sampling Period	Control	Electrolytic Treatment	Electrolytic Treatment vs. Control
*Eikenella corrodens*	1	−0.98	−2.73	−1.53
	2	−0.58	−3.20	−0.66
*Fusobacterium nucleatum*	1	1.09	−4.48	−1.09
	2	−1.95	−5.64	−3.32
*Parvimonas micra*	1	−4.22	−1.78	5.36
	2	−2.65	−2.69	2.88
*Campylobacter rectus*	1	0.21	−1.02	−4.26
	2	3.67	−0.07	−6.77

**Table 5 animals-14-03067-t005:** Analysis of variations among clinical parameters of dog oral health. Oral health benefited more from electrolytic treatment than control treatment in terms of the calculus index parameter, while for the parameters of gingival and plaque indices, changes came after regular toothbrushing.

Clinical Parameter	Source of Variation	% of Total Variation	*p* Value
Gingival index	Interaction	2.607	0.1224
	Sampling period	6.542	0.0083
	Toothbrush type	1.837	0.4836
	Dogs’ individual variation	64.60	<0.0001
Calculus index	Interaction	0.5939	0.5007
	Sampling period	1.907	0.1442
	Toothbrush type	19.70	0.0276
	Dogs’ individual variation	61.70	<0.0001
Plaque index	Interaction	0.6316	0.6618
	Sampling period	19.78	<0.0001
	Toothbrush type	3.012	0.3064
	Dogs’ individual variation	48.91	0.0005
Bleeding on probing	Interaction	2.142	0.4902
	Sampling period	3.043	0.3573
	Toothbrush type	2.874	0.2705
	Dogs’ individual variation	40.03	0.1433

**Table 6 animals-14-03067-t006:** Analysis of variations among bacterial abundance. For *E. corrodens*, *F. nucleatum*, and *P. micra*, the greater variations in the results were related to the toothbrush type rather than the sampling period.

Bacteria	Source of Variation	% of Total Variation	*p* Value
*E. corrodens*	Interaction	2.953	0.2058
	Sampling period	1.688	0.3871
	Toothbrush type	22.07	0.0057
	Dogs’ individual variation	40.34	0.0093
*F. nucleatum*	Interaction	1.154	0.5612
	Sampling period	7.075	0.0436
	Toothbrush type	10.19	0.0690
	Dogs’ individual variation	45.99	0.0058
*P. micra*	Interaction	1.853	0.2545
	Sampling period	6.497	0.0222
	Toothbrush type	15.36	0.0491
	Dogs’ individual variation	54.21	<0.0001
*C. rectus*	Interaction	3.950	0.1410
	Sampling period	4.960	0.1017
	Toothbrush type	0.5541	0.8010
	Dogs’ individual variation	74.01	<0.0001

## Data Availability

Data are contained within the article and Appendix A.

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
