# Peer review of "The Effects of Electrolytic Technology Toothbrush Application on the Clinical Parameters and Bacteria Associated with Periodontal Disease in Dogs"

_animals, 2024, doi:10.3390/ani14213067_

Round 1
Reviewer 1 Report
Comments and Suggestions for Authors
16- reword: Periodontal disease in dogs is a common and worldwide problem. As written does not make any sense.
17- Doesn’t make grammatical sense as written.
18- What medical condition?
18- change reflected to observed
18- may be observed as an unpleasant odor
19- …which can lead to the loss of teeth
19- Furthermore, it (what is “it”)
19- change was also to is
20- add the before heart
21- delete therefore
21- Re-word the rest of this sentence such as “ In order to prevent this problem and enhance the quality of life in dogs, tooth brushing is recommended”
21-24- reword, confusing and run on sentence
25-26- Reword- such as “ In order to elevate the effectiveness of tooth brushing, ionic electric toothbrushes with electrolytic technology are recommended as part of a dog’s oral hygiene regimen”
This manuscript needs to be thoroughly reviewed for readability. I stopped correcting manuscript for readability.
Title- I would not use the term electrolytic and non-electrolytic- I would use the term electric and non-electric.
17- 90% needs a reference
21- needs a reference
21-24-needs reference
42-43- There are not two types of periodontal disease, there are 4 stages. Gingivitis is the first stage followed by gingivitis and the varying levels of attachment loss. All stages of periodontal disease have gingivitis.
50-51- needs reference
74-75- needs reworded as edema, hypertrophy, and ulceration, etc., or not related to color.
80-82- needs reference
83-85 needs to be moved up around line 75 as it is discussing gingiva and gingival inflammation
85- This sentence does not make sense. What is a heavier clinical case.
92- reference
92-96-reference
98- change electrolytic to electric
101-103 I don’t understand what this means. Please reword
103-104 what is the grip and what is the brittle? Please define or reword
106-107- reword
128- examinate is a word- examinated is not a word. Please replace with a more appropriate word. The word examinated is used multiple time.
135- I do not recommend brushing of deciduous teeth. When the teeth are resorbing as would be expected, brushing can be uncomfortable. This would then potentially have the dog associate oral homecare (brushing) with pain.
149- All the patients should have started off with clean oral cavities. Beginning with oral cavities that have calculus present is a flaw design of the study.
128-130 says that the dogs were examined by a veterinary dental specialist, but does not include whether they were anesthetized. Additionally when given bleeding upon probing scores, one would assume that this would be performed with the patient under general anesthesia, yet there is no mention of general anesthesia.
Table 2- where did you get the index scores? It would be best to cite the scores and to have an already established grading scale.
288-289- reference
300-303- need to specify that this is in humans, not animals
309- apical should be subgingival
The study population is biased as it included animals whose owners were already involved in tooth brushing and dental home care of some sort prior to the research. Additionally, the study did not describe anesthesia of the patients to collect samples and to assess bleeding upon probing scores which should have been performed under general anesthesia.
Comments on the Quality of English Language
After consulting with the assistant editor (Mr. Vanja Bašić), I discontinued editing the manuscript for grammar and readability. This manuscript needs to be thoroughly reviewed for readability.
Overall, I cannot stress enough the poor readability of this manuscript. It needs to be evaluated for proper grammar, run-on sentences and over usage of filler words.
Author Response
Dear reviewer,
We have extensively rewritten our manuscript. According to your and other reviewers suggestions, we have been able to see, check and correct the most of the commented issues. In further responses we addressed the problem, we have stated the references you asked and we have changed the “false sense” text which comes from writing in nonnative language. Large portion of abstract, short summary and introduction is now rewritten, correcting suggested irregularities.
The Authors

Reviewer 2 Report
Comments and Suggestions for Authors
This is an interesting article to compare the effect of electrolytic and nonelectrolytic toothbrush on oral health in dogs.
Abstract: The aim is too long and there is no need to contain the methodology. On the other hand, The methods part is too short with missing of details. Results reported no data. Its far too general and look like conclusion. The conclusion should rewrite to address the aim (which need to rewrite and more succinct)
Introduction: It is long-winded. The background should shows the important of the ptpic and highlight the research gap. Rewrite the aim of the study.
Method: The Tables are poorly presented. Table I presenting raw data with no synthesis. Table 2 should be reformatted and in a single page. Table 3 to 5 are incomplete. As a general rule, table along with its title and the footnotes, should provide enough information so that a reader can determine what the table is showing without having to look for additional information in the text of the article. The last two columns in Table 6 and 7 are unnecessary. The charts in Figure 1 are too small with no statistics.
Discussion. The first three paragraphs are not discussion and should be summarised in Introduction.
Conclusion should address the aim, and remove unrelated statements.
Thsi paper needs copy editing
Author Response
Dear Reviewer,
Firstly, we would like to thank you for your compliment and comments. We tried to give adequate answers to your suggestions.
The Authors
Abstract: The aim is too long and there is no need to contain the methodology. On the other hand, The methods part is too short with missing details. Results reported no data. It’s far too general and look like conclusion. The conclusion should rewrite to address the aim (which need to rewrite and more succinct)
Thank you. We have rewritten the whole simple summary and the abstract part.
Simple Summary: There is a lack of data about usage of electrolytic toothbrushes and their effect on overall oral health in dogs, however there have been previous favorable reports on usage in human dentistry. The effect of electrolytic toothbrush of novel design was investigated on 26 dogs of various breeds in 8-week study with evening toothbrushing for 2 minutes. The daily usage of electrolytic toothbrush improves the oral health clinical parameters and reduces abundance of some of the bacteria associated with periodontal disease in dogs. The encouraging results support usage of electrolytic tooth brush in routine hygiene practice.
Abstract: Aim of this study was to compare the effect of electrolytic and nonelectrolytic electric toothbrush on dogs oral health and the presence of the common bacteria associated with periodontal disease. The problem of periodontal disease in dogs is common and worldwide distributed. The toothbrushing procedure is advised to prevent periodontal disease with additional benefit if electrolytic toothbrushes are used in dog oral hygiene. The 26 dogs enrolled the eight-week study divided in two groups – treatment and control. Daily toothbrushing was performed on all dogs using the same dog toothbrush, with control group power source disengaged. Oral examination was conducted on the 0th, 4th and 8th week of the study with sampling for bacterial analysis. The study was designed as blind for owners, veterinarian and laboratory. Improvement of average gingival index (from 0.55 to 0.31) and calculus index (from 0.55 to 0.38) in the treatment group was recorded. In the control group, after initial improvement of plaque index (from 0.97 to 0.53) on the 8th week it significantly rose to 1.21 (p<0.05). Relative bacterial abundance revealed reduction in all four tested bacteria in treatment group, while in control group the Campylobacter rectus levels rose by 3.67 log2 compared the 0th and the 8th week. No adverse effects have been recorded in both groups.
Introduction: It is long-winded. The background should show the important of the topic and highlight the research gap. Rewrite the aim of the study.
Thank you for your observation, we have shortened the introduction, we see that is not custom to add subtitles in introduction but we made separate paragraphs about: periodontal disease, clinical evaluation, pathogenic bacteria, treatment and control, electrolytic technology and purpose and aim and purpose of the study to make more reader friendly text.
Method: The Tables are poorly presented. Table I presenting raw data with no synthesis. Table 2 should be reformatted and in a single page. Table 3 to 5 are incomplete. As a general rule, table along with its title and the footnotes, should provide enough information so that a reader can determine what the table is showing without having to look for additional information in the text of the article. The last two columns in Table 6 and 7 are unnecessary. The charts in Figure 1 are too small with no statistics.
Thank you, we have added more extensive table and figures footnotes, we also removed last two columns of Tables 6 and 7. We added an asterisk for p<0.05 where applicable. Size and placing of Table 2 and Figures question have been sent to the Editor.
We further have the answer from the Editor that this will be addressed by layout team during the production: “For the questions regarding the positioning of figures and tables, do as you see fit and give your explanation to the reviewers based on that. If there are problems with figures and tables later on our layout team will help take care of it.”
Discussion. The first three paragraphs are not discussion and should be summarised in Introduction.
Thank you, those paragraphs have been moved to introduction, but afterward introduction part is heavily rewritten.
Conclusion should address the aim, and remove unrelated statements.
Thank you, the conclusion has been upgraded.

Reviewer 3 Report
Comments and Suggestions for Authors
The article, titled "The Electrolytic Technology Toothbrush Beneficial Effect in Dog Oral Hygiene," presents insights into the use of electrolytic technology for dog oral care, but several areas need improvement.
1.The title suggests a clear benefit of the electrolytic toothbrush, but the evidence does not fully support this conclusion. Additionally, as studies on this technology already exist, the article should focus more on the specific results and novel findings it contributes, rather than introducing the technology.
2. Study design and population: A key issue is the heterogeneous dog population, including various breeds with different oral health conditions. Brushing was performed by owners, introducing variability in technique and duration, which undermines the reliability of the results. A more controlled study environment is necessary to draw more accurate conclusions.
3. Efficacy markets, bacterial findings, and protocol: The article lacks clarity regarding the specific efficacy markers used. It would benefit from a stronger focus on the novelty of its findings, particularly in terms of any new bacteria identified, comparisons of bacterial load with other studies, and specific changes in oral health scoring. A protocol summary at the beginning of the materials and methods section—covering brushing frequency, sampling procedures, and how these scores were calculated—would significantly improve the reader’s understanding.
4.Dietary and envinronmental factors: While the study controlled for diet continuity, it did not standardize food types, meal frequency, or water type—factors that can significantly impact oral health. These variables should have been considered or recorded to provide a more accurate assessment of the toothbrush’s effectiveness.
5. Financial and practical considerations: Given that the electrolytic toothbrush showed results similar to regular brushing, the article should address the cost-benefit ratio, particularly from a clinician’s perspective. Since the technology likely incurs higher costs, the financial implications and whether the marginal differences justify the expense should be discussed.
Future studies maybe can compare the electrolytic toothbrush with non-mechanical alternatives, such as antiseptic rinses, to determine if this technology offers a significant advantage. Moreover, the study should focus on the novelty of its results, emphasizing any new findings in bacterial load or specific oral health improvements.
In summary, while the study provides useful data, a more focused title, clarification of methodology, emphasis on novel findings, and consideration of financial and practical implications would enhance its contribution to the field.
Author Response
Dear reviewer,
Firstly, we would like to thank you for your comments, as they have significantly contributed to improving our manuscript. Some of your remarks prompted valuable discussions among the authors, encouraging us to reconsider certain aspects, and, as a result, we have gained valuable insights from your feedback. We have made every effort to give comprehensive answers to your suggestions.
The Authors

Round 2
Reviewer 1 Report
Comments and Suggestions for Authors
Simple summary; needs to be reworded. There are missing words to make complete sentences. It is difficult to read and understand.
Assuming that you are using an electrolytic toothbrush and not an electric toothbrush, the differences between the two toothbrushes need to be described as well as the potential benefits of each. This should be in the introduction. This would be best served by describing an electric toothbrush and more definitively describing the electrolytic toothbrush around 160-190.
46 redundant word
Abstract: needs to be reworded. There are missing words to make complete sentences. It is difficult to read and understand.
52- conscious? Sedated? Anesthetized exams?
117- gingival bleeding by what measure?
Introduction: needs to be reworded. There are missing words to make complete sentences. It is difficult to read and understand due to sentence structure as well as either missing or extra words that do not make sense grammatically.
125-128 doesn’t make sense
130-134- redundant- combine
145- profusion is not the right word
Tense changes from singular to plural need to be corrected
158-159- reword
Singular and plural words need to be reviewed- e.g. teeth or tooth around 165
Around 165- multiple run-on sentences and grammar errors.
178- not sue adsorption is the correct word choice
185-190- needs reworded
193-195- needs reworded
196-199- needs reworded
213- collection is not the correct word
212- redundant word- (with)
214- reword
215- sampling what?
217- probing
218- what was sampled?
219- the previous statement clims that study patients were examined either by veterinarian or veterinary specialist, but then says they were all examined by veterinary specialist, which statement is inaccurate? Please clarify
219-221- reword
232-234- reword to make simpler
236- but is not the correct word consider the following: Both groups utilized the same toothbrush. The electric power source was disengaged for the control group.
243- remove also
245-24- this statement is confusing- option and voluntary have similar meanings
251- sentence is missing words to make grammatically correct
253- conceived is not the correct word
After this point, I stopped addressing the gramatical content. The paper still needs significant improvement in readability.
261- were they performed by a board certified veterinary dentist? A general practitioner with interest in dentistry? A doctor who specializes in dentistry is not clear to what their credentials are.
444- apical- should be subgingival
445- what methodology constraints?
445-448- this sentence does not make any sense
449-453- this does not make any sense
471-474- needs reworded
were there any anti-inflammatory medications that wee administered during the study period that may have influenced GI or BOP?
It is never clarified if the patients were anesthetized for this study. The authors commented within the cover letter regarding a protocol that is Zoletil based, but never confirmed that the animals were sedated or anesthetized. A simple statement of “The animals used for the study were anesthetized for oral examination and sample collection according to previously established protocols (18)” would suffice. One would assume they were as there is a bleeding upon probing score. This should only be acquired if a patient is anesthetized.
Comments on the Quality of English Language
Thank you for improving the grammar of the paper. I appreciate significant improvement of the test. Unfortunately, there is still needed editing of the grammar to improve the readability.
Author Response
Dear reviewer,
We accepted all of your comments. After we have corrected manuscript, and have written answers for you and after that we sent it to English editing by MDPI service.
Simple summary; needs to be reworded. There are missing words to make complete sentences. It is difficult to read and understand.
Corrected to: There is a lack of data about usage of electrolytic toothbrushes and their effect on oral health in dogs. The aim of this study was to investigate beneficial effect of electrolytic toothbrush compared to mechanical toothbrushing in everyday dogs routine. The daily usage of electrolytic toothbrush improves the oral health clinical parameters and reduces abundance of some periodontal disease associated bacteria in dogs. The encouraging results support usage of electrolytic tooth brush in routine hygiene practice
Assuming that you are using an electrolytic toothbrush and not an electric toothbrush, the differences between the two toothbrushes need to be described as well as the potential benefits of each. This should be in the introduction. This would be best served by describing an electric toothbrush and more definitively describing the electrolytic toothbrush around 160-190.
Corrected: We said that the development of oral pathological conditions, daily tooth brushing is considered a critical home care routine for dogs [8,18,19]. The use of electrolytic toothbrush should bring additional benefits regarding how it produces an effect. Then we state that the mechanical tooth brushing action I supplemented with an electrolytic effect, and then the mechanism of action of an electrolytic toothbrush described stating that the electric circuit block the cross-linking of adhesive and preventing bacterial attachment. We also state that dental plaque prevention is confirmed in human dentistry.
46 redundant word
corrected
Abstract: needs to be reworded. There are missing words to make complete sentences. It is difficult to read and understand.
52- conscious? Sedated? Anesthetized exams?
corrected
117- gingival bleeding by what measure?
Removed.
Introduction: needs to be reworded. There are missing words to make complete sentences. It is difficult to read and understand due to sentence structure as well as either missing or extra words that do not make sense grammatically.
125-128 doesn’t make sense
Rewritten
Periodontal disease has been reported in correlation with general health status concerning renal, hepatic, cardiac disease as well as detrimental blood health parameters [5,9]
130-134- redundant- combine
The state of gingival health is assessed by various indexes that are designed to give a numerical value to clinical status [10]. In brief, the gingival index (GI) measures the inflammation level, based on the status of the gingival margin and clinical symptoms such as redness, edema, hypertrophy, bleeding and ulceration
145- profusion is not the right word
Corrected to proliferation
Tense changes from singular to plural need to be corrected
158-159- reword
Reworded to: To prevent the development of oral pathological conditions, daily tooth brushing is considered as a critical home care routine for dogs [8,18,19]
Singular and plural words need to be reviewed- e.g. teeth or tooth around 165
Around 165- multiple run-on sentences and grammar errors.
Although two similarly charged surfaces should electrostatically repel each other, in realistic conditions plaques stick to teeth, because positively charged calcium (Ca2+) ion in the saliva reduces the natural negative charge of both bacteria and teeth surface allowing the bacteria and the teeth surface to get closer together, where, van der Waals forces, which are effective at very short distances, allow the bacteria to stick to the teeth [21].
178- not sue adsorption is the correct word choice
Changed to attachment
185-190- needs reworded
Reworded to: The development of dog toothbrushes capable of efficiently and effectively removing dental plaques led to electrolytic technology, previously recognized in human dentistry [20], beneficial in patients with multibracket appliances as well [23].
193-195- needs reworded
The zoonotic impact of dog periodontal disease is noticeable for pet owners due to similarities between the conditions in dogs and humans, furthermore the canine animal model is considered to be of the essence for human periodontal disease studies [22,24]
196-199- needs reworded
The aim of this study was to investigate whether the use of the of electrolytic toothbrush in dogs, would be beneficial compared to nonelectrolytic and to see if it could be used in everyday routine oral health prevention practice. To conduct the analysis, dogs’ the oral health was examined the levels of the prominent periodontitis related bacteria were monitored.
213- collection is not the correct word
212- redundant word- (with)
corrected
214- reword
Enrolment criteria included that none of the dogs have had received antibiotics treatment in the month period prior to the study. No dogs received antibacterial or anti-inflammatory medicines during the study.
215- sampling what?
Removed
217- probing
corrected
218- what was sampled?
Added gingival swabs
219- the previous statement clims that study patients were examined either by veterinarian or veterinary specialist, but then says they were all examined by veterinary specialist, which statement is inaccurate? Please clarify
Thank you, we wanted to say that all dogs were checked both on general health and then the oral exam
Rewritten to : The dogs were veterinary examined, the study included dogs in generally healthy condition, with Gingival index, Plaque index and Bleeding on Probing scores no worse than 2.
219-221- reword
Rewritten to: The owners were asked to withdraw from the study if any signs of dogs discomfort related to tooth brushing procedure.
232-234- reword to make simpler
The experiment was designed as the blind study (neither the owners, veterinary dentist nor the swab examiners knew which group dog belonged to). All dogs were randomly distributed to the two groups.
236- but is not the correct word consider the following: Both groups utilized the same toothbrush. The electric power source was disengaged for the control group.
Corrected as recommended Both groups utilized the same toothbrush. The electric power source was disengaged for the control group (C), while electrolytic treatment group (ET) had fully functional tootbrush.
243- remove also
removed
245-24- this statement is confusing- option and voluntary have similar meanings
Corrected: There was also an optional dogs’ oral health observation in next two months after the experiment (data not shown).
251- sentence is missing words to make grammatically correct
Corrected: Toothbrush is made of rubbery soft substance allowing the dogs to chew on it and not to break if bitten.
253- conceived is not the correct word
Changed to: set
After this point, I stopped addressing the gramatical content. The paper still needs significant improvement in readability.
261- were they performed by a board certified veterinary dentist? A general practitioner with interest in dentistry? A doctor who specializes in dentistry is not clear to what their credentials are.
This would be the local classification of post-graduation studies, similar to medical specialty in human medicine.
Changed to: veterinary dentist
444- apical- should be subgingival
Changed to subgingival, thank you
445- what methodology constraints?
Corrected: The observation protocol revealed minor issues in sensitivity. For example, the dog with deposits of plaque or calculus on one tooth received the same score as the dog with the same amount of deposits on multiple teeth. We found this to be methodology constraint, so the results might be considered as the worst-case scenario.
445-448- this sentence does not make any sense
Corrected to: We found this to be methodology constraint, so the results might be considered as the worst-case scenario.
449-453- this does not make any sense
Corrected to: The dogs’ CI improved from electrolytic treatment more than the control, while GI improvement comes after subsequent regular toothbrushing more than from the toothbrush choice. The BOP hasn’t changed notably in the individual dogs throughout the study.
471-474- needs reworded
Corrected to: The results at the 8th week in both clinical and molecular analysis were better than ones on the 4th week in both clinical and molecular analysis, and the 4th week results were better than the start of the study indicating that even the short or irregular toothbrushing would be better than no treatment at all, which is consistent with previous findings [5].
were there any anti-inflammatory medications that wee administered during the study period that may have influenced GI or BOP?
This is added in the materials and methods section, in 2.2 Study population and exclusion criteria
It is never clarified if the patients were anesthetized for this study. The authors commented within the cover letter regarding a protocol that is Zoletil based, but never confirmed that the animals were sedated or anesthetized. A simple statement of “The animals used for the study were anesthetized for oral examination and sample collection according to previously established protocols (18)” would suffice. One would assume they were as there is a bleeding upon probing score. This should only be acquired if a patient is anesthetized.
We added explicitly that all procedures were done in Zoletil based anesthesia now, one of the authors opposes stating the anesthesia as it was not the focus of the study.
Comments on the Quality of English Language
Thank you for improving the grammar of the paper. I appreciate significant improvement of the test. Unfortunately, there is still needed editing of the grammar to improve the readability.
Thank you for all your comments and corrections,
Kind regards,
The authors

Reviewer 2 Report
Comments and Suggestions for Authors
The paper can be accepted after professional editing.
Comments on the Quality of English LanguageThe paper can be accepted after professional editing. I suggested MDPI provide discount to authors for their professional editing service.
Author Response
Dear reviewer,
After corrections suggested by other reviewers, we sent manuscript to English editing by MDPI service.
Kind regards,
Authors
